# Universal approximations of permutation invariant/equivariant functions by deep neural networks

## Abstract

In this paper, we develop a theory about the relationship between $G$-invariant/equivariant functions and deep neural networks for finite group $G$. Especially, for a given $G$-invariant/equivariant function, we construct its universal approximator by deep neural network whose layers equip $G$-actions and each affine transformations are $G$-equivariant/invariant. Due to representation theory, we can show that this approximator has far fewer free parameters than usual models.

## 1 Introduction

Deep neural networks have great success in many applications such as image recognition, speech recognition, natural language process and others as Alex et al. (2012), Goodfellow et al. (2013), Wan et al. (2013), and Silver et al. (2017). A common strategy in their works is to construct larger and deeper networks. However, one of the main obstructions about using very deep and large networks for learning tasks is the so-called *curse of dimensionality*. Namely, if the parameters' dimension increase, so does the required sample size. Then, the computational complexity becomes higher. An idea to overcome this is to design models with respect to the objective structure.

Zaheer et al. (2017) designed a model adapted to machine learning tasks defined on sets, which are, from a mathematical point of view, permutation invariant or equivariant tasks. They demonstrate surprisingly good applicability on their method on population statistic estimation, point cloud classification, set expansion, and outliers detection. Empirically speaking, their results are really significant. Many researchers studied invariant/equivariant networks as Qi et al. (2017), Hartford et al. (2018), Risi Kondor (2018), Maron et al. (2019a), Bloem-Reddy & Teh (2019), Kondor & Trivedi (2018), and so on. Nevertheless, theoretical guarantee of their methods is not sufficiently considered. One of our motivations is to establish a theoretical guarantee. In this paper, we prove an *invariant/equivariant version* of the universal approximation theorem by constructing an approximator. For the symmetric group, our approximator is close to the equivariant model of Zaheer et al. (2017) in a sense (see a remark after Theorem 2.2). We can calculate the number of the free parameters appearing in our invariant/equivariant model, and show that this number is far fewer than one of the usual models.

For usual deep neural networks, a *universal approximation theorem* was first proved by Cybenko (1989). It states that, when the width goes to infinity, a (usual) neural network with a single hidden layer can, with arbitrary accuracy, approximate any continuous function with compact support. Though his theorem was only for sigmoid activation functions, there are further versions of this theorem which allows some wider classes of activation functions. In the recent literature, the most commonly used activation function is the *ReLU (Rectified Linear Unit)* function, which is the one we focus on in this paper. Some important previous works on universal approximation theorem for *ReLU* activation function are by Barron (1994), Hornik et al. (1989), Funahashi (1989), Kůrková & Sanguineti (2002), and Sonoda & Murata (2017). In particular, for a part of the proof of our main theorem, we borrow the results of Sonoda & Murata (2017) and Hanin & Sellke (2017). The interest of the universal approximation theorem in learning theory is to guarantee that *we can search in the space which contains the solutions*. The universal approximation theorem states the existence of the model which approximates the target function in arbitrary accuracy. This means that if we use the suitable algorithm, we have the desired solutions. We cannot guarantee such situations

without the universal approximation theorem. Our universal approximation theorem allows us to apply representation theory. By this point of view, we can calculate the number of free parameters of our approximator.

In the equivariant case, an essential key point of the proof is *the one to one correspondence* between $G$-equivariant functions and $\mathrm{Stab}_G(i)$-invariant functions. Here, $G$ is a finite group and $\mathrm{Stab}_G(i)$ is the subgroup of $G$ consisting of the elements which fix $i$. We first confirm this correspondence at the function level. After that, we rephrase it by deep neural networks. This correspondence enables us to reduce the equivariant case to the invariant case.

The invariant case has already established by some researchers Zaheer et al. (2017), Yarotsky (2018), Maron et al. (2019b). For $G = S_n$, here $S_n$ is the symmetric group of degree $n$, Zaheer et al. (2017) showed that a representation theorem of $S_n$-invariant function which is famous as a solution for the Hilbert's 13th problem by Kolmogorov (1956) and Arnold (1957) gives us an explicit description. Due to this theorem and the usual universal approximation theorem, we can construct a concrete deep neural network of the invariant model. Recently, Maron et al. (2019b) proved an invariant version of the universal approximation theorem for any finite group $G$ using tensor structures. We borrow their results to obtain our main results.

## 1.1 CONTRIBUTIONS

Our contributions are summarized as follows:

• We prove an invariant/equivariant version of the approximation theorems, which is a one step to understand the behavior of deep neural networks with permutations or more generally group actions.

• Using representation theory, we calculate the number of free parameters appearing in our models. As a result, the number of parameter in our models is far fewer than the one of the usual models. This means that our models are easier to train than the usual models.

• Although our model is slightly different from the equivariant model of Zaheer et al. (2017) for $G = S_n$, our theorem guarantees that our model for finite group $G$ can approximate any $G$-invariant/equivariant functions.

## 1.2 RELATED WORKS

Group theory, or symmetry is an important concept in mathematics, physics, and machine learning. In machine learning, deep symmetry networks (symnets) is designed by Gens & Domingos (2014) as a generalization of convnets that forms feature maps over arbitrary symmetry groups. Group equivariant Convolutional Neural Networks (G-CNNs) is designed by Cohen & Welling (2016), as a natural generalization of convolutional neural networks that reduces sample complexity by exploiting symmetries. The models for permutation invariant/equivariant tasks are designed by Zaheer et al. (2017) to give great results on population statistic estimation, point cloud classification, set expansion, and outlier detection.

The universal approximation theorem is one of the most classical mathematical theorems of neural networks. As we saw in the introduction, Cybenko (1989) proved this theorem in 1989 for sigmoid activation functions. After his achievement, some researchers showed similar results to generalize the sigmoid function to a larger class of activation functions as Barron (1994), Hornik et al. (1989), Funahashi (1989), Kůrková (1992) and Sonoda & Murata (2017).

As mentioned above, the invariant case has been established. For $G = S_n$, Zaheer et al. (2017) essentially proved an invariant version of an universal approximation theorem. Yarotsky (2018) gave a more explicit $S_n$-invariant approximator by a shallow deep neural network. Maron et al. (2019b) considered a $G$-invariant model with some tensor structures for any finite group $G$. An equivariant version for finite group $G$ by shallow (hyper-)graph neural networks is proved by Keriven & Peyré (2019). Our architecture of approximator is different from theirs. Moreover, although they proved only for "squashing functions" which exclude ReLU functions, our theorem allows us to use the ReLU functions. We also remark that our setting in this paper is quite general. In particular, ours include tensor structures, hence graph neural networks. It must be interesting to compare the numbers of free parameters of models of us and Keriven & Peyré (2019).

## 2 PRELIMINARIES AND MAIN RESULTS

In this paper, we treat fully connected deep neural networks. We mainly consider ReLU activation functions. Here, the ReLU activation function is defined by

$$\text{ReLU}(x) = \max(0, x).$$

We remark that our argument during this paper works for any activation functions which satisfy a usual universal approximation theorem. A deep neural network is built by stacking the blocks which consist of a linear map and a ReLU activation. More formally, it is a function $Z_i$ from $\mathbb{R}^{d_i}$ to $\mathbb{R}^{d_{i+1}}$ defined by $Z_i(\boldsymbol{x}) = \text{ReLU}(W_i \boldsymbol{x} + \boldsymbol{b}_i)$, where $W_i \in \mathbb{R}^{d_{i+1} \times d_i}$, $\boldsymbol{b}_i \in \mathbb{R}^{d_{i+1}}$. In this case, $d_i$ is called the width of the $i$-th layer. The output of the deep neural networks is

$$Y(\boldsymbol{x}) = Z_H \circ Z_{H-1} \ldots Z_2 \circ Z_1(\boldsymbol{x}), \tag{1}$$

where $H$ is called the depth of the deep neural network. We define the width of a deep neural network as the maximum of the widths of all layers.

Our main objects are deep neural networks which are invariant/equivariant with actions by a finite group $G$. We review some facts about groups and these actions here. Some more details are written in Appendix A. Let $S_n$ be the group consisting of permutations of $n$ elements $\{1, 2, \ldots, n\}$. This $S_n$ is called the symmetric group of degree $n$. The symmetric group $S_n$ acts on $\{1, 2, \ldots, n\}$ by the permutation $i \mapsto \sigma^{-1}(i)$ for $\sigma \in S_n$. By Proposition A.1 that any finite group $G$ can be regarded as a subgroup of $S_n$ for a positive integer $n$. Then, $G$ also acts on $\{1, \ldots, n\}$ by the action as an element of $S_n$. For $i \in \{1, 2, \ldots, n\}$, we define the orbit of $i$ as $O_i = G(i) = \{\sigma(i) \mid \sigma \in G\}$. Then, the set $\{1, 2, \ldots, n\}$ can be divided to a disjoint union of the orbits: $\{1, 2, \ldots, n\} = \bigsqcup_{j=1}^{m} O_{i_j}$.

Let $G$ be a finite group action on $\{1, 2, \ldots, n\}$. For $i = 1, 2, \ldots, n$, we define the stabilizer subgroup $\text{Stab}_G(i)$ of $G$ associated with $x$ by the subgroup of elements of $G$ fixing $i$. Then, by Proposition A.2, the orbit $O_i$ and the set of cosets $G/\text{Stab}_G(i)$ are bijective. When $G = S_n$, $X = \{1, 2, \ldots, n\}$, and $x = 1$, we set $\text{Stab}_n(1) = \text{Stab}(1) := \text{Stab}_G(x)$.

We next consider an action of $S_n$ on the vector space $\mathbb{R}^n$. The left action "$\cdot$" of $S_n$ on $\mathbb{R}^n$ is defined by

$$\sigma \cdot \boldsymbol{x} = \sigma \cdot (x_1, x_2, \ldots, x_n)^\top = (x_{\sigma^{-1}(1)}, x_{\sigma^{-1}(2)}, \ldots, x_{\sigma^{-1}(n)})^\top$$

for $\sigma \in S_n$ and $\boldsymbol{x} = (x_1, \ldots, x_n)^\top \in \mathbb{R}^n$. We also call this the permutation action of $S_n$ on $\mathbb{R}^n$. If there is an injective group homomorphism $\varphi \colon G \hookrightarrow S_n$, $G$ acts on $\mathbb{R}^n$ by the permutation action as an element of $S_n$ by $\varphi$: For $\sigma \in G$ and $\boldsymbol{x} \in \mathbb{R}^n$, we define $\sigma \cdot \boldsymbol{x} := \varphi(\sigma) \cdot \boldsymbol{x}$. Then, we simply say that $G$ acts on $\mathbb{R}^n$.

**Example 2.1.** The finite group $G = S_2$ is embedded into $S_3$ by $\varphi \colon (1\ 2) \mapsto (1\ 2)$, where $(i\ j)$ is transposition between $i$ and $j$. In this case, the orbit decomposition of $\{1, 2, 3\}$ is $\{1, 2\} \sqcup \{3\}$. By this embedding $\varphi$, an $S_2$-action on $\mathbb{R}^3$ is defined: $\varphi((1\ 2)) \cdot (x_1, x_2, x_3)^\top = (x_2, x_1, x_3)^\top$.

**Example 2.2** (Tensors). The group action of $G$ which is a subgroup of $S_n$ on tensors as in Maron et al. (2019b) is realized as follows: An $S_n$-action on $\mathbb{R}^{n^k \times a}$ is defined by the following injective homomorphism $\varphi \colon G \hookrightarrow S_{an^k}$: We fix a bijection from $\{1, \ldots, an^k\}$ to $\{1, \ldots, n\}^k \times \{1, 2, \ldots, a\}$, and for $\sigma \in S_n$, $\varphi(\sigma) \in S_{an^k}$ is defined by

$$\varphi(\sigma) \cdot (i_1, \ldots, i_k, j) = (\sigma^{-1}(i_1), \ldots, \sigma^{-1}(i_k), j)$$

for $(i_1, \ldots, i_k) \in \{1, \ldots, n\}^k$ and $j \in \{1, 2, \ldots, a\}$. Then, for a tensor $X = (x_{i_1, \ldots, i_k, j})_{i_1, \ldots, i_k = 1 \ldots, n,\ j = 1, 2, \ldots, a} \in \mathbb{R}^{n^k \times a}$, $\sigma \in S_n$ acts on $\mathbb{R}^{n^k \times a}$ by $\varphi(\sigma) \cdot X = (x_{\sigma^{-1}(i_1), \ldots, \sigma^{-1}(i_k), j})$. This action is same as one of Maron et al. (2019b).

**Example 2.3** ($n$-tuple of $D$-dimensional vectors). We identify $\{1, 2, \ldots, nD\}$ with $\{(i, j) \mid i = 1, \ldots, n, j = 1, \ldots, D\}$ and define $\varphi \colon S_n \hookrightarrow S_{nD}$ as $\varphi(\sigma) \cdot (i, j) = (\sigma^{-1}(i), j)$. Let $(\boldsymbol{x}_1, \ldots, \boldsymbol{x}_n) \in (\mathbb{R}^D)^n$ be an $n$-tuple of $D$-dimensional vectors. Then, for $\sigma \in S_n$, $\phi(\sigma) \cdot (\boldsymbol{x}_1, \ldots, \boldsymbol{x}_n) = (\boldsymbol{x}_{\sigma^{-1}(1)}, \ldots, \boldsymbol{x}_{\sigma^{-1}(n)})$. This means a permutation of $n$ vectors.

**Definition 2.1.** Let $G$ be a finite group. We assume that an injective homomorphisms $\varphi \colon G \hookrightarrow S_m$ are given. Then, $G$ acts on $\mathbb{R}^m$. We say that a map $f \colon \mathbb{R}^m \to \mathbb{R}^n$ is $G$-*invariant* if $f(\varphi(\sigma) \cdot \boldsymbol{x}) = f(\boldsymbol{x})$ for any $\sigma \in G$ and any $\boldsymbol{x} \in \mathbb{R}^m$. We also assume that $\psi \colon G \hookrightarrow S_n$. Then, we say that a map $f \colon \mathbb{R}^m \to \mathbb{R}^n$ $G$-*equivariant* if $f(\varphi(\sigma) \cdot \boldsymbol{x}) = \psi(\sigma) \cdot f(\boldsymbol{x})$ for any $\sigma \in G$.

When $G = S_n$ and the actions are induced by permutation, we call $G$-invariant (resp. $G$-equivariant) functions as *permutation invariant* (resp. *permutation equivariant*) functions.

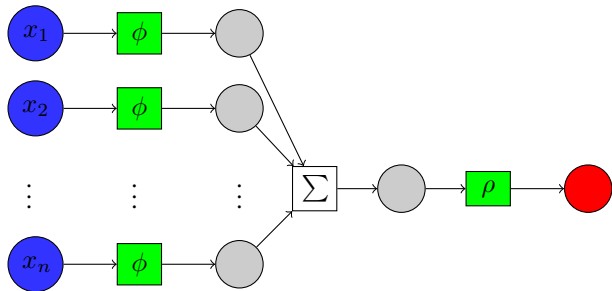

Diagram 1: A neural network approximating $S_n$-invariant function $f$. In blue: the inputs, in red: the output, in green: $\rho$ and $\phi$ who have to be learned.

We define $G$-invariance and $G$-equivariance for deep neural networks. We can easily confirm that the models in Zaheer et al. (2017) satisfies these properties.

**Definition 2.2.** We say that a deep neural network $Z_H \circ Z_{H-1} \ldots Z_2 \circ Z_1$ as (1) is *G-equivariant* if an action of $G$ on each of layers $\mathbb{R}^{d_i}$ is given by embedding $G \hookrightarrow S_{d_i}$ and the all corresponding map $Z_i \colon \mathbb{R}^{d_i} \to \mathbb{R}^{d_{i+1}}$ is $G$-equivariant. We say that a deep neural network is *G-invariant* if there is a positive integer $c \leq H$ such that $G$-actions on each layer $\mathbb{R}^{d_i}$ for $1 \leq i \leq c+1$ are given and the corresponding map $Z_i \colon \mathbb{R}^{d_i} \to \mathbb{R}^{d_{i+1}}$ is $G$-equivariant for $1 \leq i \leq c-1$ and the map $Z_c \colon \mathbb{R}^{d_c} \to \mathbb{R}^{d_{c+1}}$ is $G$-invariant.

Some approximation theorems for invariant functions have been already known:

**Proposition 2.1** (*G*-invariant version of universal approximation theorem)**.** *Let $G$ be a finite group which is a subgroup of $S_n$. Let $K$ be a compact set in $\mathbb{R}^n$ which is stable for the corresponding $G$-action in $\mathbb{R}^n$. Then, for any $f \colon K \to \mathbb{R}^m$ which is continuous and $G$-invariant and for any $\epsilon > 0$, the following $G$-invariant ReLU neural networks $\mathcal{N}_G^{\mathrm{inv}} \colon \mathbb{R}^n \to \mathbb{R}^m$ satisfy that these represented functions $R_{\mathcal{N}_G^{\mathrm{inv}}}$ satisfy $\|f - R_{\mathcal{N}_G^{\mathrm{inv}}}\|_\infty < \epsilon$:*

- *$\mathcal{N}_G^{\mathrm{inv}} = \Sigma \circ L_H \circ \mathrm{ReLU} \circ \cdots \circ L_1$, where $\Sigma$ is the summation ($G$-invariant part) and $L_i \colon \mathbb{R}^{n^{d_i} \times a_i} \to \mathbb{R}^{n^{d_{i+1}} \times a_{i+1}}$ is a linear map such that $L_i(g \cdot X) = g \cdot L_i(X)$ for any $g \in G$ and any $X \in \mathbb{R}^{n^{d_i} \times a_i}$. Here, the actions on each layers except for the output layer are same as Example 2.2.*

- *For $G = S_n$. $\mathcal{N}_G^{\mathrm{inv}} = \mathcal{N}_\rho \circ \Sigma \circ (\mathcal{N}_\phi, \ldots, \mathcal{N}_\phi)^\top$, where $\mathcal{N}_\rho$ (resp. $\mathcal{N}_\phi$) is a deep neural network approximating $\rho$ (resp. $\phi$) defined below.*

This proposition for $G = S_n$ is proven by Zaheer et al. (2017), Yarotsky (2018). For general finite group $G$, Maron et al. (2019b) proved it. Diagram 1 illustrates the $S_n$-invariant ReLU neural network appeared in Proposition 2.1. The key ingredient of the proof by Zaheer et al. (2017) is the following Kolmogorov-Arnold representation theorem:

**Theorem 2.1** (Zaheer et al. (2017) Kolmogorov-Arnold's representation theorem for permutation actions)**.** *Let $K \subset \mathbb{R}^n$ be a compact set. Then, any continuous $S_n$-invariant function $f \colon K \to \mathbb{R}$ can be represented as*

$$f(x_1, \ldots, x_n) = \rho\left(\sum_{i=1}^n \phi(x_i)\right) \tag{2}$$

*for some continuous function $\rho \colon \mathbb{R}^{n+1} \to \mathbb{R}$. Here, $\phi \colon \mathbb{R} \to \mathbb{R}^{n+1}; x \mapsto (1, x, x^2, \ldots, x^n)^\top$.*

Since $\phi(x)$ has only one variable, we line up the copies of the network which approximates $\phi(x)$. Then, by combining $\Sigma$ and the network which approximates $\rho$, we obtain the network which approximates $f$. By using the theorem of Hanin & Sellke (2017) (resp. Sonoda & Murata (2017)), we obtain the bound of the width (resp. the depth) for approximation of $\phi$ and $\rho$. Maron et al. (2019b) proved this proposition using a tensor structure.

The main theorem is a $G$-equivariant version of universal approximation theorem. To state the main theorem, we need some notation. Let $G$ be a finite group acting on $\mathbb{R}^n$. We set the orbit decomposition

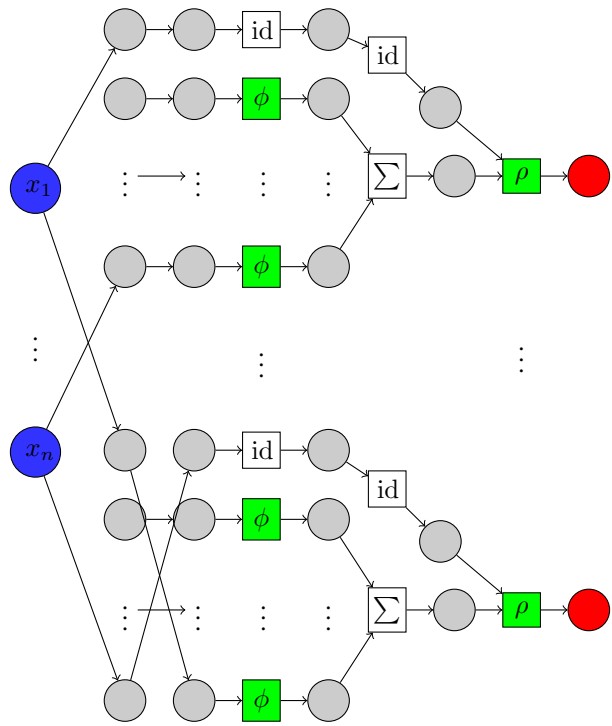

Diagram 2: A neural network approximating $S_n$-equivariant map $F$

of $\{1, 2, \ldots, n\}$ as $\{1, 2, \ldots, n\} = \bigsqcup_{j=1}^{m} O_{i_j}$, and let $O_{i_j} = \{i_{j1} = i_j, i_{j2}, \ldots, i_{jl_j}\}$. Without loss of generality, we may reorder $\{1, 2, \ldots, n\}$ as

$$O_{i_1} = \{1, 2, \ldots, l_1\}, O_{i_2} = \{l_1 + 1, l_1 + 2, \ldots l_1 + l_2\}, \ldots$$

and $i_j = \sum_{k=1}^{j-1} l_k + 1$ for $j = 1, 2, \ldots, m$. For each $j$, let $G = \bigsqcup_{k=1}^{l_j} \mathrm{Stab}_G(i_j)\tau_{j,k}$ be the coset decomposition by $\mathrm{Stab}_G(i_j)$. Then, we may assume that $\tau_{j,k} \in G$ satisfies $\tau_{j,k}^{-1}(i_j) = i_j + k$ for $k = 1, 2, \ldots, l_j$. Then, the main theorem is the following:

**Theorem 2.2** ($G$-equivariant version of universal approximation theorem). *Let $G$ be a finite group which is a subgroup of $S_n$. Let $K$ be a compact set in $\mathbb{R}^n$ which is stable for the corresponding $G$-action in $\mathbb{R}^n$. Then, for any $F \colon K \to \mathbb{R}^n; \boldsymbol{x} \mapsto (f_1, \ldots, f_n)$ which is continuous and $G$-equivariant and for any $\epsilon > 0$, the following $G$-invariant ReLU neural network $\mathcal{N}_G^{\mathrm{equiv}} \colon \mathbb{R}^n \to \mathbb{R}^n$ satisfies that these represented functions $R_{\mathcal{N}_G^{\mathrm{equiv}}}$ satisfy $\|F - R_{\mathcal{N}_G^{\mathrm{equiv}}}\|_\infty < \epsilon$:*

$$\mathcal{N}_G^{\mathrm{equiv}} = (\mathcal{N}_{\mathrm{Stab}_G(i_1)}^{\mathrm{inv}} \circ \tau_{1,1}, \ \mathcal{N}_{\mathrm{Stab}_G(i_1)}^{\mathrm{inv}} \circ \tau_{1,2}, \ \ldots, \ \mathcal{N}_{\mathrm{Stab}_G(i_m)}^{\mathrm{inv}} \circ \tau_{m,l_m})^\top.$$

*Here, $\mathcal{N}_{\mathrm{Stab}_G(i_j)}^{\mathrm{inv}}$ is the $\mathrm{Stab}_G(i_j)$-invariant deep neural network approximating $f_{i_j}$ as in Proposition 2.1, and the actions on each layers are defined as follows: Each of hidden layers are written by $\mathbb{R}^{n^2} \otimes_{\mathbb{R}} V$ for a vector space $V$. On this space, $\sigma \in G$ acts on $(\boldsymbol{x}_1, \ldots, \boldsymbol{x}_n) \otimes v \in \mathbb{R}^{n^2} \otimes_{\mathbb{R}} V$ ($\boldsymbol{x}_i \in \mathbb{R}^n$) by*

$$\sigma \cdot ((\boldsymbol{x}_1, \ldots, \boldsymbol{x}_n) \otimes v) = (\widetilde{\sigma}_{1,1} \cdot \boldsymbol{x}_{\sigma^{-1}(1)}, \ldots, \widetilde{\sigma}_{m,l_m} \cdot \boldsymbol{x}_{\sigma^{-1}(n)}) \otimes v,$$

*where $\widetilde{\sigma}_{j,k}$ is the element of $\mathrm{Stab}_G(i_j)$ satisfying $\widetilde{\sigma}_{j,k} = \tau_{j,k'} \sigma \tau_{j,k}^{-1}$ for some $k' = 1, 2, \ldots, l_j$.*

For $G = S_n$, when $\mathcal{N}_G^{\mathrm{equiv}}$ is represented by $Z_H \circ Z_{H-1} \circ \cdots \circ Z_1$ as in (1), we can consider that $Z_2, \ldots, Z_H$ are $S_n$-equivariant by usual permutation. In this sense, our model is close to the $S_n$-equivariant model in Zaheer et al. (2017).

Our strategy for the proof is the following: At first, we establish the correspondence between $\mathrm{Stab}_G(i_j)$-invariant functions and $G$-equivariant functions. By this correspondence, we take

$\mathrm{Stab}_G(i_j)$-invariant function $f$ corresponding to the objective function $F$. By Proposition 2.1, we can approximate $f$ by a $\mathrm{Stab}_G(i_j)$-invariant network $\mathcal{N}^{\mathrm{inv}}_{\mathrm{Stab}_G(i_j)}$. Using $\mathcal{N}^{\mathrm{inv}}_{\mathrm{Stab}_G(i_j)}$, we construct the $G$-equivariant network which approximates $F$. Diagram 2 illustrates the $S_n$-equivariant ReLU neural network appeared in Theorem 2.2.

Due to our universal approximation theorems, if the free parameters of the invariant/equivariant models are fewer than the ones of the usual models, we have a guarantee for using the invariant/equivariant models. The following definition illustrates the swapping of nodes.

**Definition 2.3.** Let $X = \{1, \ldots, M\}$ be an index set of nodes in a layer. We say an $S_n$-action on $X$ is a union of permutations if $X = \bigsqcup X_i$, where each $X_i$ has $n$ elements and $S_n$ acts on $X_i$ by permutation.

**Theorem 2.3.** *Let $\mathcal{N}$ be an $S_n$-invariant model of depth $D$ and width $M$ whose number of the equivariant layers is $d$ (resp. an $S_n$-equivariant model of depth $D$ and width $M$). Assume that the action is a union of permutations on nodes in each equivariant layer. Then, the number of free parameters in this model is bounded by $M^{2D} \cdot (2/n^2)^d$ (resp. $M^{2D} \cdot (2/n^2)^D$).*

Note that the number of free parameters in the fully connected model is $M^{2D}$. Hence, this theorem implies that *the free parameters of the invariant/equivariant models are far fewer than the ones of the usual models.*

## 3 EQUIVARIANT CASE

In this section, we prove Theorem 2.2, namely, the equivariant version of the universal approximation theorem. The key ingredient is the following proposition (proof is in Appendix C):

**Proposition 3.1.** *Notations are same as Theorem 2.2. Then, a map $F\colon \mathbb{R}^n \to \mathbb{R}^n$ is $G$-equivariant if and only if $F$ can be represented by $F = (f_1 \circ \tau_{1,1}, f_1 \circ \tau_{1,2}, \ldots, f_m \circ \tau_{m,l_m})^\top$ for some $\mathrm{Stab}_G(i_j)$-invariant functions $f_j\colon \mathbb{R}^n \to \mathbb{R}$. Here, $\tau_{j,k} \in G$ is regarded as a linear map $\mathbb{R}^n \to \mathbb{R}^n$.*

For simplicity, we prove Theorem 2.2 only for $G = S_n$. We can show the general case by a similar argument. More precisely, we construct an $S_n$-equivariant deep neural network approximating the given $S_n$-equivariant function. Similarly, we can prove this theorem for any finite group $G$. To show Theorem 2.2 for $G = S_n$, we divide the proof to four steps as follows:

1. By Proposition 3.1 proved below, we reduce the argument on $S_n$-equivariant map $F$ to the one of $\mathrm{Stab}(1)$-invariant function $f$.

2. Modifying Theorem 2.1, we have a representation of $\mathrm{Stab}(1)$-invariant function $f$.

3. Using the above representation, we have a $\mathrm{Stab}(1)$-invariant deep neural net which approximates $f$ and construct a deep neural network approximating $F$.

4. We introduce a certain action of $S_n$ on $(\mathbb{R}^n)^n$ which appears the first hidden layer naturally and show the $S_n$-equivariance between the input layer and the first hidden layer.

We first investigate step 1. We recall that, during this section, we only consider the action of $S_n$ on $\mathbb{R}^n$ induced from permutation $\sigma \cdot (x_1, \ldots, x_n)^\top = (x_{\sigma^{-1}(1)}, \ldots, x_{\sigma^{-1}(n)})^\top$. Then, we remark that the orbit of 1 by this action is the total set $O_1 = \{1, 2, \ldots, n\}$ and the coset decomposition of $S_n$ by $\mathrm{Stab}(1)$ is $S_n = \bigsqcup_{i=1}^n \mathrm{Stab}(1)(1\,i)$. Thus, we have the following:

**Corollary 3.1.** *A map $F\colon \mathbb{R}^n \to \mathbb{R}^n$ is $S_n$-equivariant if and only if there is $\mathrm{Stab}(1)$-invariant function $f\colon \mathbb{R}^n \to \mathbb{R}$ satisfying $F = (f \circ (1\,1), f_1 \circ (1\,2), \ldots, f_m \circ (1\,n))^\top$. Here, $(i\,j) \in S_n$ is the transposition between $i$ and $j$ and is regarded as a linear map $\mathbb{R}^n \to \mathbb{R}^n$.*

Next, we consider step 2. The stabilizer subgroup $\mathrm{Stab}_n(1)$ is isomorphic to $S_{n-1}$ as a group by Lemma. Hence, we can regard the $\mathrm{Stab}(1)$-invariant function $f\colon \mathbb{R}^n \to \mathbb{R}$ as an $S_{n-1}$-invariant function. This point of view allows us to apply Theorem 2.1 to $f$. Hence, we have the following representation theorem of $\mathrm{Stab}(1)$-invariant functions as a corollary of Theorem 2.1.

**Corollary 3.2** (Representation of $\mathrm{Stab}(1)$-invariant function)**.** *Let $K \subset \mathbb{R}^n$ be a compact set, let $f\colon K \longrightarrow \mathbb{R}$ be a continuous and $\mathrm{Stab}(1)$-invariant function. Then, $f(\boldsymbol{x})$ can be represented as*

$$f(\boldsymbol{x}) = f(x_1, \ldots, x_n) = \rho\left(x_1, \sum_{i=2}^n \phi(x_i)\right),$$

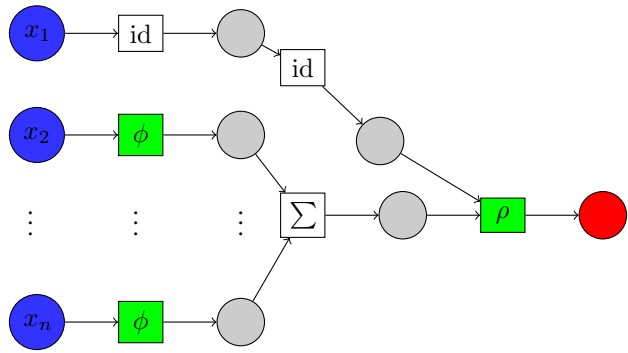

Diagram 3: A neural network approximating the $\mathrm{Stab}(1)$-invariant function $f$

*for some continuous function $\rho\colon \mathbb{R}^{n+1} \longrightarrow \mathbb{R}$. Here, $\phi\colon \mathbb{R} \to \mathbb{R}^n$ is similar as in Theorem 2.1.*

By this corollary, we can represent the $\mathrm{Stab}(1)$-invariant function $f\colon \mathbb{R}^n \longmapsto \mathbb{R}$ as $f = \rho \circ L \circ \Phi$, where $\Phi \colon \mathbb{R}^n \to \mathbb{R} \times (\mathbb{R}^n)^{n-1}$ and $L \colon \mathbb{R} \times (\mathbb{R}^n)^{n-1} \to \mathbb{R} \times \mathbb{R}^n$ are

$$\Phi(x_1, \ldots, x_n) = (x_1, \phi(x_2), \ldots, \phi(x_n)), \quad L(x, (\boldsymbol{y}_1, \ldots, \boldsymbol{y}_{n-1})) = \left(x, \sum_{i=1}^{n-1} \boldsymbol{y}_i \right).$$

Then, we consider step 3, namely, the existence of $\mathrm{Stab}(1)$-invariant deep neural network approximating the function $f$. After that, using this approximator, we construct a deep neural network approximating $S_n$-equivariant function $F$. By a slight modification of the invariant version of Proposition 2.1 for $\mathrm{Stab}(1)$-invariant case, there exists a sequence of deep neural networks $\{A_m\}_m$ (resp. $\{B_m\}_m$) which converges to $\Phi$ (resp. $\rho$) uniformly. Then, the sequence of deep neural networks $\{B_m \circ L \circ A_m\}_m$ converges to $f = \rho \circ L \circ \Phi$ uniformly.

Now, $f$ can be approached by the following deep neural network by replacing $\rho$ and $\Phi$ by universal approximators as Diagram 3. We remark that the left part (the part of before taking sum) of this deep neural network is naturally equivariant for the action of $\mathrm{Stab}(1)$. For an $S_n$-equivariant map $F : \mathbb{R}^n \to \mathbb{R}^n$ with the natural action, by Proposition 3.1, there is a unique $\mathrm{Stab}(1)$-invariant function $f$ such that $F(\boldsymbol{x})_i = (f \circ (1\ i))(\boldsymbol{x})$. Here, $F(\boldsymbol{x}) = (F(\boldsymbol{x})_1, \ldots, F(\boldsymbol{x})_n)^\top$ and we regard any element of $S_n$ as a map from $\mathbb{R}^n$ to $\mathbb{R}^n$. By the argument in this section, we can approximate $f$ by the previous deep neural network $\{B_m \circ L \circ A_m\}_m$. Substituting $B_m \circ L \circ A_m$ for $f$, we construct a deep neural network approximating $F$ as Diagram 2.

The represented function of this neural network of $F_i$ is $B_m \circ L \circ A_m \circ (1\ i)$. The map $F$ splits into two parts, the part of transpositions and part of $(f, f, \ldots, f)^\top$. On the deep neural network corresponding to $F$ as above, the latter part corresponds to the layers from the first hidden layer to the output layer. This part is the $n$ copies of same $\mathrm{Stab}(1)$-invariant deep neural network (an approximation of $(f, f, \ldots, f)^\top$). Thus, this part is clearly made of equivariant stacking layers for the permutation action of $S_n$. Hence, it is remained to show that the former part is also $S_n$-equivariant.

We here investigate bounds of the width and the depth of approximators. By Theorem 2.1, each of $\phi$ and $\rho$ can be approximated by a shallow neural network. Hence, if we do not bound the width, we can obtain deep neural network approximating $F$ with depth three. On the other hand, by Theorem 2.1 again, if we do not bound the depth, $\phi$ (resp. $\rho$) can be approximated by a deep neural network with width $n + 1$ (resp. $n + 2$). Thus, we can obtain a deep neural network approximating $F$ with width bounded from above by $n^3$.

Finally, as step 4, we show that our deep neural network is an $S_n$-equivariant deep neural network. The most difficult part is to show the equivariance between the input layer $\mathbb{R}^n$ and the first hidden layer $\mathbb{R}^{n^2}$ presented by a function $g\colon \mathbb{R}^n \to \mathbb{R}^{n^2}$ as $g = (g_1, \ldots, g_n)^\top$ for $g_i = \mathrm{ReLU} \circ l \circ (1\ i)$ for a certain $\mathrm{Stab}(1)$-invariant linear function $l\colon \mathbb{R}^n \to V$. (Although $V$ is equal to $\mathbb{R}^n$, we distinguish them to stress the difference.) Unfortunately, the permutation action on the latter space $\mathbb{R}^{n^2}$ does not make the function $g$ equivariant. For this reason, we need to define another action of $S_n$ on $\mathbb{R}^{n^2}$ exploiting the $\mathrm{Stab}(1)$-equivariance among each copies.

**Definition 3.1.** We suppose that $\mathrm{Stab}(1)$ acts on $\mathbb{R}^n$ by permutation, denoted by $\sigma \cdot \boldsymbol{x}$ (i.e., by regarding $\mathrm{Stab}(1)$ as a subgroup of $S_n$). Then, we define the action "$*$" of $S_n$ on $\mathbb{R}^{n^2}$ as follows:

$$\sigma * (\boldsymbol{x}_1, \ldots, \boldsymbol{x}_n) = (\tilde{\sigma}_1 \cdot \boldsymbol{x}_{\sigma^{-1}(1)}, \ldots, \tilde{\sigma}_n \cdot \boldsymbol{x}_{\sigma^{-1}(n)}) = (x_{\tilde{\sigma}_j^{-1}(i), \sigma^{-1}(j)})_{i,j=1,\ldots,n}$$

for any $\sigma \in S_n$, and for any $(\boldsymbol{x}_1, \boldsymbol{x}_2, \ldots, \boldsymbol{x}_n) = (x_{i,j})_{i,j=1,\ldots,n} \in (\mathbb{R}^n)^n$. Here, for any $i$, $\tilde{\sigma}_i$ is an element of $\mathrm{Stab}(1)$ defined as $(1\ i)\sigma = \tilde{\sigma}_i(1\ \sigma^{-1}(i))$.

We will prove the well-definedness of this action "$*$" in Appendix D. This action is obtained by the injective homomorphism

$$\varphi \colon S_n \hookrightarrow S_{n^2}; \ \sigma \mapsto \varphi(\sigma), \ \ \varphi(\sigma) \cdot (i,j) = (\tilde{\sigma}_j^{-1}(i), \sigma^{-1}(j))$$

for $(i,j) \in \{1, 2, \ldots, n\}^2$. We remark that this action "$*$" naturally appears in representation theory as the induced representation, which is an operation to construct a representation of group $S_n$ from a representation of the subgroup $\mathrm{Stab}(1)$ of $S_n$.

We conclude the proof of Theorem 2.2 by showing the $S_n$-equivariance of $g$:

**Lemma 3.1.** *The function* $g \colon \mathbb{R}^n \to \mathbb{R}^{n^2}$ *is* $S_n$*-equivariant.*

The detail of proof is in Appendix E. By this lemma, we conclude the proof of Theorem 2.2. We remark that the affine transformation $g$ is corresponding to $Z_1$ in the notation (1). By a representation theoretic aspect, this is an intertwining operator between these representation spaces. This affine transformation has $n^3$ free parameters a priori. However, by $S_n$-equivariance and a representation theoretic argument, in principle, $g$ can be written by only five parameters. By a similar argument, for the other hidden layers, the affine transformation $Z_i : \mathbb{R}^{n^2} \otimes V_i \to \mathbb{R}^{n^2} \otimes V_{i+1}$ has $15 \dim V_i \dim V_{i+1}$ parameters (though $n^4 \dim V_i \dim V_{i+1}$ a priori).

## 4 DIMENSION REDUCTION

In this section, we give the proof of Theorem 2.3.

**Proposition 4.1.** *Let* $Z_l = \mathrm{ReLU} \circ W_l \colon \mathbb{R}^M \to \mathbb{R}^N$ *be an* $S_n$*-equivariant map. Assume that the* $S_n$*-action on* $\mathbb{R}^M$ *and* $\mathbb{R}^N$ *is a union of permutations. Then,* $n$ *divides* $M$ *and* $N$*, and the number of the free parameters in* $W_l$ *is equal to* $2MN/n^2$.

*Proof.* Since $\mathbb{R}^M$ and $\mathbb{R}^N$ have union of permutation actions, by considering the orbit of the coordinates, we see that $n$ divides $M$ and $N$. Let us write $\mathbb{R}^M = (\mathbb{R}^n)^{M'}$ and $\mathbb{R}^N = (\mathbb{R}^n)^{N'}$. In this case, $W_l$ is written by sum of $n \times n$ matrices $V_{ij}$, namely $W_l = (V_{ij})_{i,j=1}^n$. Here, each $n \times n$ matrix $V_{ij}$ corresponds to the linear map:

$$\mathbb{R}^n \hookrightarrow (\mathbb{R}^n)^{M'} \xrightarrow{Z_l} (\mathbb{R}^n)^{N'} \twoheadrightarrow \mathbb{R}^n,$$

where the first map is the inclusion to coordinates of $(\mathbb{R}^n)^{M'}$ and the last map is projection to coordinates of $(\mathbb{R}^n)^{N'}$. Since these constructions are taken to be compatible with $S_n$-action, we see that $\mathrm{ReLU} \circ V_{ij}$ is $S_n$-equivariant. If the activation functions are bijective, we are done because of the same discussion as in the proof of Lemma 3 in Zaheer et al. (2017). But in our case, we need more discussion, which is in Appendix G. $\square$

*Proof of Theorem 2.3.* By Proposition 4.1, the number of the free parameter in each equivariant layer is bounded by $2M^2/n^2$. Hence we obtain the desired bound. $\square$

## 5 CONCLUSION

We introduced some invariant/equivariant models of deep neural networks which are universal approximators for invariant/equivariant functions. The universal approximation theorems in this paper and the discussion in Section 4 show that although the free parameters of our invariant/equivariant models are far fewer than the ones of the usual models, the invariant/equivariant models can approximate the invariant/equivariant functions to arbitrary accuracy. Our theory also implies that there is much possibility that the group models behave as the usual models for the tasks related to groups and representation theory can be powerful tool for theory of machine learning. This must be a good perspective to develop the models in deep learning.

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

# A  REVIEW OF GROUPS AND GROUP ACTIONS

Let $G$ be a set with product, i.e., for any $g, h \in G$, $gh$ is defined an element of $G$. Then, $G$ is called a group if $G$ satisfies the following conditions:

1. There is an element $e \in G$ such that $eg = ge = x$ for all $g \in G$.
2. For any $g \in G$, there is an element $g^{-1} \in G$ such that $gg^{-1} = g^{-1}g = e$.
3. For any $g, h, i \in G$, $(gh)i = g(hi)$ holds.

Let $G, G'$ be two finite groups. We say a map $\varphi \colon G \to G'$ is a (group) homomorphism if $\varphi(gh) = \varphi(g)\varphi(h)$. This means that the map $\varphi$ preserves the group structures of $G$ in $G'$.

Next, we review actions of groups. Let $X$ be a set. An action of $G$ (or $G$-action) on $X$ is defined as a map $G \times X \to X; (g, x) \mapsto g \cdot x$ satisfying the following:

1. For any $x \in X$, $e \cdot x = x$.
2. For any $g, h \in G$ and $x \in X$, $(gh) \cdot x = g \cdot (h \cdot x)$.

If these conditions are satisfied, we say that $G$ acts on $X$ by a left action.

**Example A.1.** An example which we mainly consider in this paper is the permutation group $S_n$ of $n$ elements:

$$S_n = \{\sigma \colon \{1, \ldots, n\} \to \{1, \ldots, n\}; \text{bijective}\}$$

and the product of $\sigma, \tau \in S_n$ is given by the composition $\sigma \circ \tau$ as maps. $S_n$ acts on the set $\{1, 2, \ldots, n\}$ by the permutation $\sigma \cdot i = \sigma^{-1}(i)$.

We remark that actions of $S_n$ on $X$ is not unique, for example, the trivial action $\sigma \cdot x = x$ for any $\sigma \in S_n, x \in X$ is also one of action. Hence, when we stress the difference of some actions, we use some distinguished notation for each actions as "$\cdot$" or "$*$" etc.

Let $G$ be a group and $H$ a subset of $G$. We call $H$ a subgroup of $G$ if $H$ is a group with the same product as $G$.

**Example A.2.** Let $G$ be a finite group acting on a set $X$. For an element $x \in X$, the stabilizer subgroup $\mathrm{Stab}_G(x)$ of $G$ with respects to $\{x\}$ is defined by

$$\mathrm{Stab}_G(x) = \{g \in G \mid g \cdot x = x\}.$$

When $G = S_n$ and $x = 1$, we use the following notation: $\mathrm{Stab}_n(1) = \mathrm{Stab}(1) := \mathrm{Stab}_G(x)$.

Let $G, G'$ be two groups. If there is an injective homomorphism $\varphi \colon G \to G'$, the image $\varphi(G) \subset G'$ can be a subgroup of $G'$. Then, we say that $\varphi$ is an embedding of group $G$ to $G'$. Moreover, if $G'$ acts on a set $X$, then $G$ also acts on $G'$ through $\varphi$, i.e., by $\varphi(\sigma) \cdot x$ for $\sigma \in G$ and $x \in X$.

Then, the following proposition holds:

**Proposition A.1.** *Any finite group $G$ can be embedded into $S_n$ for some $n$.*

*Proof.* Let $n := |G|$ and $G = \{g_1, g_2, \ldots, g_n\}$. For any $g \in G$, $gg_i = g_j$ for some $j \in \{1, 2, \ldots, n\}$ as $gg_i \in G$. We set $\sigma_g^{-1}(i) = j$. Then, we define $\sigma \colon G \to S_n; g \mapsto \sigma_g$. It is easy to show that this $\sigma$ is an injective homomorphism. $\square$

This proposition implies that any finite group $G$ can be realized as a permutation action on $\mathbb{R}^n$ for some $n$.

Let $G$ be a finite group acting on $X$. Then, for $x \in X$, we define the ($G$-)orbit $O_x$ of $x$ as

$$O_x = G \cdot x = \{g \cdot x \mid g \in G\}.$$

Then, for $x, y \in X$, the relation that $x$ and $y$ are in a same orbit is an equivalent relation. Hence, $X$ can be divided to a disjoint union of the equivalent classes of this equivalent relation:

$$X = \bigsqcup_{j=1}^{m} O_{x_j}.$$

We call this the ($G$-)orbit decomposition of $X$.

Let $H$ be a subgroup of a finite group $G$. Then, for $g \in G$, the set

$$gH = \{gh \in G \mid h \in H\}$$

is called the left coset of $H$ with respect to $g$. The relation that two elements $g$ and $g'$ are in a same coset is also an equivalent relation. Hence, we can divide $G$ to a disjoint union of equivalent classes of this relation:

$$G = \bigsqcup_{i=1}^{l} g_i H.$$

We call this decomposition the right coset decomposition of $G$ by $H$. We set $G/H$ as the set of the left cosets of $G$ by H:

$$G/H = \{g_1 H, g_2 H, \ldots, g_l H\}.$$

Then, there is a relation between an orbit and a set of cosets:

**Proposition A.2.** *Let $G$ be a finite group acting on a set $X$. For $x \in X$, the map*

$$G/\mathrm{Stab}_G(x) \to O_x;\ g\mathrm{Stab}_G(x) \to g \cdot x$$

*is bijective.*

*Proof.* It is easy to check well-definedness and bijectivity. $\qquad\square$

For $G = S_n$ acting on $\{1, 2, \ldots, n\}$, the $G$-orbit of $1$ is only one, i.e., $O_1 = \{1, 2, \ldots, n\}$. Hence, the following holds (the permutation action of $S_n$ is defined by taking inverse $\sigma^{-1}$, hence we need to consider the set of right cosets):

**Corollary A.1.** *The map*

$$\mathrm{Stab}_n(1)\backslash S_n \to \{1, 2, \ldots, n\}; \mathrm{Stab}_n(1)\sigma \mapsto \sigma^{-1}(1)$$

*is bijective.*

## B   PROOF OF PROPOSITION 2.1

*Proof of Proposition 2.1.* We may assume $N = 1$. In fact, since we consider the $L^\infty$-norm, if all components of $\|f - R_\mathcal{N}\|$ is bounded by $\epsilon$, then $\|f - R_\mathcal{N}\|_\infty \le \epsilon$ holds. For $N = 1$, we have $f\colon K \to \mathbb{R}$. Then, by Theorem 2.1, we obtain the representation of $f$ as

$$f(x_1, \ldots, x_n) = \rho\left(\sum_{i=1}^{n} \phi(x_i)\right).$$

By the universal approximation property of ReLU deep neural network, we can find two sequences of ReLU deep neural network $\{\mathcal{N}_k^\rho\}_k$ and $\{\mathcal{N}_k^\phi\}_k$ such that their corresponding functions $\{F_{\mathcal{N}_k^\rho}\}_k$ and $\{F_{\mathcal{N}_k^\phi}\}_k$ tend to $\rho$ and $\phi$ for the $L^\infty$-norm when $k$ tends to infinity. Let $\{\mathcal{N}_k\}_k$ be the sequence of networks whose corresponding functions are $F_{\mathcal{N}_k}\colon (x_1, \ldots, x_n) \mapsto \left(F_{\mathcal{N}_k^\rho}\left(\sum_{i=1}^{n} F_{\mathcal{N}_k^\phi}(x_i)\right)\right)$. To show that $\{F_{\mathcal{N}_k}\}_k$ uniformly tends to $f$, we use the following lemma:

**Lemma B.1.** *If $\{f_k\}_k$ tends uniformly to $f$, $\{g_k\}_k$ tends uniformly to $g$ and $f$ is uniformly continuous, then $\{f_k \circ g_k\}_k$ tends uniformly to $f \circ g$.*

*proof of Lemma B.1.* For any $k$, we have

$$\begin{aligned}
|f_k \circ g_k(x) - f \circ g(x)| &= |f_k \circ g_k(x) - f \circ g_k(x) + f \circ g_k(x) - f \circ g(x)| \\
&\le \underbrace{|f_k \circ g_k(x) - f \circ g_k(x)|}_{\le \|f_k - f\|_\infty} + |f \circ g_k(x) - f \circ g(x)|.
\end{aligned}$$

Let $\epsilon > 0$. By the uniform continuity of $f$, there is a $\delta > 0$ such that for all $x, y$ satisfying $|x - y| \le \delta$, $|f(x) - f(y)| \le \epsilon/2$ holds. Then for large enough $k$, we have both $\|g_k - g\|_\infty \le \delta$ which implies that for any $x$, $|f \circ g_k(x) - f \circ g(x)| \le \epsilon/2$, and $\|f_k - f\|_\infty \le \epsilon/2$. Hence, for any $k$ large enough, $\|f_k \circ g_k - f \circ g\| \le \epsilon$ holds. $\qquad\square$

Now using Lemma B.1, we have that $(x_1, \ldots, x_n) \mapsto \sum_{i=1}^n F_{\mathcal{N}_k^\phi}(x_i)$ tends to $(x_1, \ldots, x_n) \mapsto \sum_{i=1}^n \phi(x_i)$, because $(x_1, \ldots, x_n) \mapsto \sum_{i=1}^n x_i$ is Lipschitz (by triangular inequality) so uniformly continuous. Then, using Lemma B.1 again, we obtain the result, since $\rho$ is continuous on a compact set so uniformly continuous. Moreover, by Sonoda & Murata (2017), we can approximate $\phi$ and $\rho$ by shallow networks. Hence, we have an approximation by a deep neural network $\mathcal{N}$ which has two hidden layers and the width is not bounded. By Hanin & Sellke (2017), we can respectively approximate each of $\phi$ and $\rho$ by some neural networks whose width are bounded by $n + 2$ and the depth is not bounded. Hence, we have an approximation by a deep neural network $\mathcal{N}$ whose width is bounded by $n(n + 2)$ and the depth is not bounded.

The invariant function $f$ is approached by a deep neural network $\mathcal{N}$ having the following diagram:

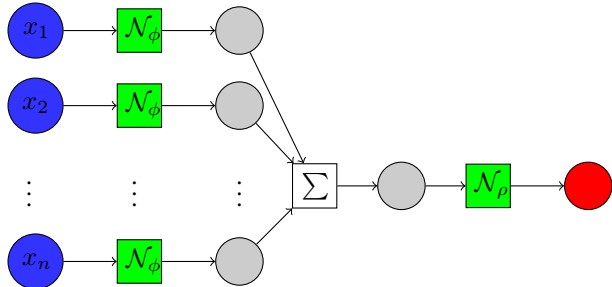

Let us show that this is a $S_n$-*invariant* deep neural network. Since the sum ($\Sigma$) is an invariant function, we can divide this neural network in two parts : the fist part on the left of the symbol $\Sigma$ and the second on the right. For each layer $\mathbb{R}^{d_i}$ of $\mathcal{N}_\phi$ there is a corresponding map $Z_i : \mathbb{R}^{d_i} \to \mathbb{R}^{d_{i+1}}$. Then for each layer $(\mathbb{R}^{d_i})^n$ of the first part of $\mathcal{N}$, there is a $S_n$ action $\sigma \cdot (x_i)_i = (x_{\sigma^{-1}(i)})_i$ for $\boldsymbol{x} = (x_i)_i \in \mathbb{R})^{d_i}$, and the corresponding map $(Z_1, \cdots, Z_n) : (\mathbb{R}^{d_i})^n \to (\mathbb{R}^{d_{i+1}})^n$ is $S_n$-equivariant. On the second part of $\mathcal{N}$, there is no $S_n$ actions on the layers $\mathbb{R}^{d_j}$ except the trivial action. Hence, for this action, each corresponding map $Z_i : \mathbb{R}^{d_j} \to \mathbb{R}^{d_{i+j}}$ is invariant. This shows that the network $\mathcal{N}$ is $S_n$-*invariant*. $\qquad\square$

## C  PROOF OF PROPOSITION 3.1

*Proof of Proposition 3.1.* First, we show that for $\mathrm{Stab}_G(i_j)$-invariant function $f_j$, the map $F = (f_1 \circ \sigma_{1,1}, f_1 \circ \sigma_{1,2}, \ldots, f_m \circ \sigma_{m,l_m})^\top \in \mathbb{R}^n$ is $G$-equivariant. Without loss of generality, we may assume that

$$O_{i_1} = \{1, 2, \ldots, l_1\}, \ O_{i_2} = \{l_1 + 1, l_1 + 2, \ldots, l_1 + l_2\}, \ldots.$$

In particular, $i_j = \sum_{k=1}^{j-1} l_k + 1$. For each $j$, we set the coset decomposition by

$$G = \bigsqcup_{k=1}^{l_j} \mathrm{Stab}_G(i_j)\tau_{j,k},$$

where we may assume that $\tau_{j,k}$ satisfies $\tau_{j,k}^{-1}(i_j) = i_j + k$. Now, for $\sigma \in G$, there is a unique $k'$ such that $\tau_{j,k}\sigma = \widetilde{\sigma}_{j,k}\tau_{j,k'}$. Hence, the $(ij + k)$-th entry becomes

$$f_j \circ \tau_{j,k} \circ \sigma = f_j \circ \widetilde{\sigma}_{j,k} \circ \tau_{j,k'} = f_j \circ \tau_{j,k'}.$$

On the other hand, we have

$$\sigma^{-1}(i_j + k) = \tau_{j,k'}^{-1}\widetilde{\sigma}_{j,k}^{-1}\tau_{j,k}(i_j + k) = i_j + k'.$$

This shows $F$ is $G$-equivariant.

Conversely, we assume that $F \colon \mathbb{R}^n \to \mathbb{R}^n$ is $G$-equivariant. Let $F = (g_1, g_2, \ldots, g_n)^\top$. The orbit decomposition $O_{i_1}, \ldots, O_{i_m}$ is same as above. For $\sigma \in G$, by $\sigma \cdot F(\boldsymbol{x}) = F(\sigma \cdot \boldsymbol{x})$ for any $\boldsymbol{x} \in \mathbb{R}^n$, we have

$$g_i(\sigma \cdot \boldsymbol{x}) = g_{\sigma^{-1}(i)}(\boldsymbol{x}). \tag{3}$$

For simplicity, we consider only for $i = 1$. Then, $\sigma^{-1}(1) \in O_{i_1}$, thus $\sigma^{-1}(1) = 1, \ldots, l_1$. By the equation (3), for $\sigma \in \mathrm{Stab}_G(1)$, we have

$$g_1(\sigma \cdot \boldsymbol{x}) = g_1(\boldsymbol{x}),$$

as $\sigma \in \mathrm{Stab}_G(1)$ if and only if $\sigma^{-1} \in \mathrm{Stab}_G(1)$. Hence, $g_1$ is $G$-invariant. The equation (3) for $\tau_{1,k}$ $(k = 1, 2, \ldots, l_1)$ implies

$$g_1(\tau_{1,k} \cdot \boldsymbol{x}) = g_{\tau_{1,k}^{-1}(1)}(\boldsymbol{x}) = g_k(\boldsymbol{x}).$$

This completes the proof for the orbit $O_{i_1}$. By same arguments, we can prove the similar result for the other orbits.

□

# D  WELL-DEFINEDNESS OF THE ACTION "$*$".

We here show that the left action "$*$" of $S_n$ on $\mathbb{R}^{n^2}$ defined in Section 3 is well-defined, i.e., for any $\sigma, \tau \in S_n$ and any $X = (\boldsymbol{x}_1^\top, \ldots, \boldsymbol{x}_n^\top)^\top \in \mathbb{R}^{n^2}$, we have $\tau * (\sigma * X) = (\tau\sigma) * X$.

The permutation group $S_n$ is decomposed as

$$S_n = \bigsqcup_{i=1}^{n} \mathrm{Stab}(1)(1\ i).$$

For any $\sigma \in S_n$, because $\sigma(1\ \sigma^{-1}(1))$ is in $\mathrm{Stab}(1)$, this $\sigma$ is in the coset $\mathrm{Stab}(1)(1\ \sigma^{-1}(1))$. Apply this for $(1\ i)\sigma$ for $i$, $(1\ i)\sigma$ is in the coset

$$\mathrm{Stab}(1)(1\ ((1\ i)\sigma)^{-1}(1)) = \mathrm{Stab}(1)(1\ \sigma^{-1}(i)),$$

thus, we have a unique element $\tilde{\sigma}_i \in \mathrm{Stab}(1)$ such that

$$(1\ i)\sigma = \tilde{\sigma}_i(1\ \sigma^{-1}(i)). \tag{4}$$

For $\sigma, \tau \in S_n$ and $X = (\boldsymbol{x}_1^\top, \ldots, \boldsymbol{x}_n^\top)^\top \in \mathbb{R}^{n^2}$, we have

$$
\tau * (\sigma * X) = \tau * \begin{pmatrix} \tilde{\sigma}_1 \cdot x_{\sigma^{-1}(1)} \\ \vdots \\ \tilde{\sigma}_n \cdot x_{\sigma^{-1}(n)} \end{pmatrix} = \begin{pmatrix} \tilde{\tau}_1 \tilde{\sigma}_{\tau^{-1}(1)} \cdot x_{\sigma^{-1}(\tau^{-1}(1))} \\ \vdots \\ \tilde{\tau}_1 \tilde{\sigma}_{\tau^{-1}(n)} \cdot x_{\sigma^{-1}(\tau^{-1}(n))} \end{pmatrix}
$$
$$
= \begin{pmatrix} \tilde{\tau}_1 \tilde{\sigma}_{\tau^{-1}(1)} \cdot x_{(\tau\sigma)^{-1}(1)} \\ \vdots \\ \tilde{\tau}_1 \tilde{\sigma}_{\tau^{-1}(n)} \cdot x_{(\tau\sigma)^{-1}(n)} \end{pmatrix}
$$

Then, by the equation (4), $\tilde{\sigma}_i = (1\ i)\sigma(1\ \sigma^{-1}(i))$. Hence we have

$$\tilde{\tau}_i \tilde{\sigma}_{\tau^{-1}(i)} = (1\ i)\tau(1\ \tau^{-1}(i))(1\ \tau^{-1}(i))\sigma(1\ \sigma^{-1}(\tau^{-1}(i)))$$
$$= (1\ i)\tau\sigma(1\ (\tau\sigma)^{-1}(i)) = \widetilde{(\tau\sigma)}_i.$$

This relation implies

$$
\tau * (\sigma * X) = \begin{pmatrix} \tilde{\tau}_1 \tilde{\sigma}_{\tau^{-1}(1)} \cdot x_{(\tau\sigma)^{-1}(1)} \\ \vdots \\ \tilde{\tau}_1 \tilde{\sigma}_{\tau^{-1}(n)} \cdot x_{(\tau\sigma)^{-1}(n)} \end{pmatrix} = \begin{pmatrix} \widetilde{(\tau\sigma)}_1 \cdot x_{(\tau\sigma)^{-1}(1)} \\ \vdots \\ \widetilde{(\tau\sigma)}_n \cdot x_{(\tau\sigma)^{-1}(n)} \end{pmatrix} = (\tau\sigma) * X
$$

Thus, the action is well-defined.

To satisfy the hypothesis of Theorem 2.3, we need to check that the "$*$" action is free.

## E    PROOF OF LEMMA 3.1

*Proof of Lemma 3.1.* For any $i$, any $\sigma \in S_n$ and any $\boldsymbol{x} \in \mathbb{R}^n$, we have

$$(l \circ (1\ i) \circ I_n)(\sigma \cdot \boldsymbol{x}) = l(((1\ i)\sigma) \cdot \boldsymbol{x}).$$

Because for any $i$, there is a unique $\tilde{\sigma}_i \in \mathrm{Stab}(1)$ such that $(1\ i)\sigma = \tilde{\sigma}_i(1\ \sigma^{-1}(i))$ as in Definition 3.1, we have

$$l(((1\ i)\sigma) \cdot \boldsymbol{x}) = l(\tilde{\sigma}_i(1\ \sigma^{-1}(i)) \cdot \boldsymbol{x}) = \tilde{\sigma}_i \cdot l((1\ \sigma^{-1}(i)) \cdot \boldsymbol{x}).$$

The last equality is due to $\mathrm{Stab}(1)$-equivariance of $l$. On the other hand, by Definition 3.1, $i$-th entry of $\sigma * g(\boldsymbol{x})$ becomes

$$
\begin{aligned}
(\sigma * g(\boldsymbol{x}))_i &= \tilde{\sigma}_i \cdot g(\boldsymbol{x})_{\sigma^{-1}(i)} \\
&= \tilde{\sigma}_i \cdot (\mathrm{ReLU}(l((1\ \sigma^{-1}(i)) \cdot \boldsymbol{x}))).
\end{aligned}
$$

Because $\tilde{\sigma}_i \circ \mathrm{ReLU} = \mathrm{ReLU} \circ \tilde{\sigma}_i$ holds, $g$ is $S_n$-equivariant. $\qquad\square$

## F    DATA AUGMENTATION

Data augmentation is a common technique in empirical learning. In the case of the invariant/equivariant tasks, a possible augmentation is to make new samples by the acting permutation on samples. New samples are effective to the usual models, but not effective to our invariant/equivariant models. This is because in our models, all weights are symmetric under permutation actions. This means that our models learn augmented samples from a sample. By acting permutations, we can make $n!$ new samples from a sample. Hence, computational complexity is reduced to $1/n!$ times.

## G    DIMENSION REDUCTION

In this section, we give the proof of Proposition 4.1.

*Proof of Proposition 4.1.* Since the action on $\mathbb{R}^M, \mathbb{R}^N$ is a union of permutations on nodes in each equivariant layer, we can write $\mathbb{R}^M = (\mathbb{R}^n)^{M'}$ and $\mathbb{R}^N = (\mathbb{R}^n)^{N'}$. In this case, $W_i$ can be written as the following form;

$$
W_i = \begin{pmatrix}
V_{11} & V_{12} & \dots & V_{1M'} \\
V_{21} & V_{22} & \dots & V_{2M'} \\
\vdots & \vdots & \ddots & \vdots \\
V_{M'1} & V_{M'2} & \dots & V_{N'M'}
\end{pmatrix}
$$

, where $Vij$ are $n \times n$ matrices. Let us consider the following maps:

$$\mathbb{R}^n \xrightarrow{i} (\mathbb{R}^n)^{M'} \xrightarrow{Z_i} (\mathbb{R}^n)^{N'} \xrightarrow{p} \mathbb{R}^n$$

, where the first map is the inclusion to the coordinates started from the $(j-1)n$ th coordinate of $(\mathbb{R}^n)^{M'}$ ended at the $jn-1$ th coordinate of $(\mathbb{R}^n)^{M'}$ and the last map is the projection to the coordinates started from the $(i-1)n$ th coordinate of $(\mathbb{R}^n)^{N'}$ ended at the $in-1$ th coordinate of

$(\mathbb{R}^n)^{N'}$. We chase the elements of these maps as follows;

$$
p \circ Z_i \circ i \left( \begin{bmatrix} a_1 \\ a_2 \\ \vdots \\ a_n \end{bmatrix} \right) = p \circ Z_i \left( \begin{bmatrix} 0 \\ \vdots \\ 0 \\ a_1 \\ a_2 \\ \vdots \\ a_n \\ 0 \\ \vdots \\ 0 \end{bmatrix} \right) = p \left( \mathrm{ReLU} \circ \begin{pmatrix} V_{11} & V_{12} & \dots & V_{1M'} \\ V_{21} & V_{22} & \dots & V_{2M'} \\ \vdots & \vdots & \ddots & \vdots \\ V_{N'1} & V_{N'2} & \dots & V_{N'M'} \end{pmatrix} \begin{bmatrix} 0 \\ \vdots \\ 0 \\ a_1 \\ a_2 \\ \vdots \\ a_n \\ 0 \\ \vdots \\ 0 \end{bmatrix} \right)
$$

$$
= p \begin{pmatrix} \mathrm{ReLU} \circ V_{1j} \begin{bmatrix} a_1 \\ a_2 \\ \vdots \\ a_n \end{bmatrix} \\ \vdots \\ \mathrm{ReLU} \circ V_{M'j} \begin{bmatrix} a_1 \\ a_2 \\ \vdots \\ a_n \end{bmatrix} \end{pmatrix} = \mathrm{ReLU} \circ V_{ij} \begin{bmatrix} a_1 \\ a_2 \\ \vdots \\ a_n \end{bmatrix}
$$

Hence, each $n \times n$ matrix $V_{ij}$ induces a subneural network. Since these constructions are taken to be compatible with $S_n$-action, we see that $f = \mathrm{ReLU} \circ V_{ij} : \mathbb{R}^\kappa \to \mathbb{R}^\kappa$ is $S_n$-equivariant. If the activation functions are bijective, we are done because of the same discussion as in the proof of Lemma 3 in Zaheer et al. (2017). But since ReLU functions are not bijective, we need more discussion. Let us take a transposition $\sigma = (p\ q)$. Since $f$ is $S_n$ -equivariant, $\sigma \circ f(\boldsymbol{x}) = f(\sigma \circ \boldsymbol{x})$ holds for any $\boldsymbol{x}$. We have

$$
\sigma \circ f(\boldsymbol{x}) = \sigma \left( \mathrm{ReLU} \circ \begin{pmatrix} a_{11} & \dots & a_{1n} \\ \vdots & \ddots & \vdots \\ a_{p1} & \dots & a_{pn} \\ \vdots & \ddots & \vdots \\ a_{q1} & \dots & a_{qn} \\ \vdots & \ddots & \vdots \\ a_{n1} & \dots & a_{nn} \end{pmatrix} \begin{bmatrix} x_1 \\ x_2 \\ \vdots \\ x_n \end{bmatrix} \right) = \mathrm{ReLU} \left( \sigma \circ \begin{pmatrix} a_{11} & \dots & a_{1n} \\ \vdots & \ddots & \vdots \\ a_{p1} & \dots & a_{pn} \\ \vdots & \ddots & \vdots \\ a_{q1} & \dots & a_{qn} \\ \vdots & \ddots & \vdots \\ a_{n1} & \dots & a_{nn} \end{pmatrix} \begin{bmatrix} x_1 \\ x_2 \\ \vdots \\ x_n \end{bmatrix} \right)
$$

$$
= \mathrm{ReLU} \left( \sigma \circ \begin{bmatrix} \sum_{j=1}^n a_{1j}x_j \\ \vdots \\ \sum_{j=1}^n a_{pj}x_j \\ \vdots \\ \sum_{j=1}^n a_{qj}x_j \\ \vdots \\ \sum_{j=1}^n a_{nj}x_j \end{bmatrix} \right) = \mathrm{ReLU} \left( \begin{bmatrix} \sum_{j=1}^n a_{1j}x_j \\ \vdots \\ \sum_{j=1}^n a_{qj}x_j \\ \vdots \\ \sum_{j=1}^n a_{pj}x_j \\ \vdots \\ \sum_{j=1}^n a_{nj}x_j \end{bmatrix} \right) = \mathrm{ReLU} \left( \begin{pmatrix} a_{11} & \dots & a_{1n} \\ \vdots & \ddots & \vdots \\ a_{q1} & \dots & a_{qn} \\ \vdots & \ddots & \vdots \\ a_{p1} & \dots & a_{pn} \\ \vdots & \ddots & \vdots \\ a_{n1} & \dots & a_{nn} \end{pmatrix} \begin{bmatrix} x_1 \\ x_2 \\ \vdots \\ x_n \end{bmatrix} \right).
$$

On the other hand, we have

$$
f(\sigma \circ \boldsymbol{x}) = f\left(\begin{bmatrix} x_1 \\ \vdots \\ x_q \\ \vdots \\ x_p \\ \vdots \\ x_n \end{bmatrix}\right) = \mathrm{ReLU}\left(\begin{pmatrix} a_{11} & \cdots & a_{1p} & \cdots & a_{1q} & \cdots & a_{1n} \\ a_{21} & \cdots & a_{2p} & \cdots & a_{2q} & \cdots & a_{2n} \\ \vdots & \ddots & \vdots & \ddots & \vdots & \ddots & \vdots \\ a_{n1} & \cdots & a_{np} & \cdots & a_{nq} & \cdots & a_{nn} \end{pmatrix}\begin{bmatrix} x_1 \\ \vdots \\ x_q \\ \vdots \\ x_p \\ \vdots \\ x_n \end{bmatrix}\right)
$$

$$
= \mathrm{ReLU}\left(\begin{pmatrix} a_{11} & \cdots & a_{1q} & \cdots & a_{1p} & \cdots & a_{1n} \\ a_{21} & \cdots & a_{2q} & \cdots & a_{2p} & \cdots & a_{2n} \\ \vdots & \ddots & \vdots & \ddots & \vdots & \ddots & \vdots \\ a_{n1} & \cdots & a_{nq} & \cdots & a_{np} & \cdots & a_{nn} \end{pmatrix}\begin{bmatrix} x_1 \\ \vdots \\ x_p \\ \vdots \\ x_q \\ \vdots \\ x_n \end{bmatrix}\right).
$$

**Claim 1.**
$$
\begin{pmatrix} a_{11} & \cdots & a_{1q} & \cdots & a_{1p} & \cdots & a_{1n} \\ a_{21} & \cdots & a_{2q} & \cdots & a_{2p} & \cdots & a_{2n} \\ \vdots & \ddots & \vdots & \ddots & \vdots & \ddots & \vdots \\ a_{n1} & \cdots & a_{nq} & \cdots & a_{np} & \cdots & a_{nn} \end{pmatrix} = \begin{pmatrix} a_{11} & \cdots & a_{1n} \\ \vdots & \ddots & \vdots \\ a_{q1} & \cdots & a_{qn} \\ \vdots & \ddots & \vdots \\ a_{p1} & \cdots & a_{pn} \\ \vdots & \ddots & \vdots \\ a_{n1} & \cdots & a_{nn} \end{pmatrix}
$$

*Proof of Claim 1.* We have

$$
\mathrm{ReLU}\left(\begin{pmatrix} a_{11} & \cdots & a_{1n} \\ \vdots & \ddots & \vdots \\ a_{q1} & \cdots & a_{qn} \\ \vdots & \ddots & \vdots \\ a_{p1} & \cdots & a_{pn} \\ \vdots & \ddots & \vdots \\ a_{n1} & \cdots & a_{nn} \end{pmatrix}\begin{bmatrix} x_1 \\ x_2 \\ \vdots \\ x_n \end{bmatrix}\right) = \mathrm{ReLU}\left(\begin{pmatrix} a_{11} & \cdots & a_{1q} & \cdots & a_{1p} & \cdots & a_{1n} \\ a_{21} & \cdots & a_{2q} & \cdots & a_{2p} & \cdots & a_{2n} \\ \vdots & \ddots & \vdots & \ddots & \vdots & \ddots & \vdots \\ a_{n1} & \cdots & a_{nq} & \cdots & a_{np} & \cdots & a_{nn} \end{pmatrix}\begin{bmatrix} x_1 \\ \vdots \\ x_p \\ \vdots \\ x_q \\ \vdots \\ x_n \end{bmatrix}\right)
$$

for any $\boldsymbol{x}$. Hence, the the $l$-th coordinate of the left hand side is positive if and only if the one of the right hand side is positive. It is clear that each equation is positive for infinitely many $\boldsymbol{x}$. This implies that the coefficients of each equations coincide. Hence, we have the desired result. □

We show that $a_{pp} = a_{qq}$ holds for any $p, q$. We can see that the $(p\,q)$ entry of the matrix of the left hand side in the claim is equal to $a_{pp}$. Similarly, the $(p\,q)$ entry of the matrix of the right hand side in the claim is equal to $a_{qq}$. Hence, by the claim, we have $a_{pp} = a_{qq}$. We show that if $i \neq j$ and $s \neq t$, $a_{ij} = a_{st}$ holds. Consider the $(i\,q)$ entry of each matrices, the one of the left hand side is equal to $a_{ip}$ and the one of the right hand side is equal to $a_{iq}$. Hence, we have $a_{ip} = a_{iq}$, where $i \neq p$ ane $i \neq q$. By the symmetry, $a_{pi} = a_{qi}$ holds for any $i \neq p$ and $i \neq q$. Hence, we have

$$
a_{ij} = a_{sj} = a_{st}
$$

for any $i \neq j$ and $s \neq t$. Hence, we can write $V_{ij} = \lambda \boldsymbol{I} + \gamma(\boldsymbol{1}\boldsymbol{1}^\top)$. □

