# OpenReview forum: "Universal approximations of permutation invariant/equivariant functions by deep neural networks"
_ICLR.cc/2020/Conference — Reject_

### Official Review · AnonReviewer3 · 2019-10-23
**Official Blind Review #3**

**Rating:** 3

**Review:**

*Paper summary*

The authors develop a universal approximation theorem for neural networks that are symmetric with respect to the symmetric group (permutations). They also formally show that the number of free parameters is to train an equivariant network is smaller that the number in a non-equivariant network, leading to better sample complexity.

*Paper decision*

I have decided to give this paper a weak reject. The contributions are clear and the paper is written well enough for publication. That said, the significance of the contribution is not clear to me, since there are other papers in the literature doing the same.

I must admit that the subject material is out of my region of expertise, so my judgement may be a little miscalibrated.

*Supporting arguments and questions for the authors*

The paper is clearly written by people who have a firm grasp of their subject. The contributions are well laid out and the following proofs are clearly placed in the paper. In terms of constructive criticism, I think it would be helpful to readers to give an intuition behind why the contributions are necessary and to add a sense of the motivation behind them. This would open up the paper to a broader audience.

For me, something that was not clear was how exactly this work is different from Zaheer et al., (2017). It would be nice if these differences were spelled out for me. It was also not clear why it was necessary to present the contribution that with each symmetry you introduce you reduce the number of parameters exponentially? I guess the intuition behind this is clear, but I was wondering why this is necessary to spell out with a Theorem (2.3 I believe). What exactly doe this result imply?

In terms of clarity, the preliminaries section is well written and quite clear. I enjoyed the summary. That said, it is quite advanced, and it would be useful to point more novice readers to elementary texts, where they could brush up on their group theory. Furthermore, the grammar in places is a bit tenuous, but on the whole the writing is understandable.

**Experience Assessment:**

I do not know much about this area.

**Review Assessment: Checking Correctness Of Derivations And Theory:**

I did not assess the derivations or theory.

**Review Assessment: Checking Correctness Of Experiments:**

N/A

**Review Assessment: Thoroughness In Paper Reading:**

N/A

---

> ### Author Response · Authors · 2019-11-14
> **Response to Review #3**
>
> We thank you for your constructive comments.
>
> > For me, something that was not clear was how exactly this work is different from Zaheer et al., (2017). It would be nice if these differences were spelled out for me.
>
> In the paper of Zaheer et al, the group is the symmetric group $S_n$ and the action deals only with permutation. In that situation, we proposed an invariant or equiavariant DNN model with $ aI + b11^T$ as an affine map between the middle layers, and experimented with various tasks to demonstrate its performance. The paper essentially shows that the universal approximation theorem for invariant functions. However, it was not mentioned whether any equivariant function could be represented in their model. Our result means that any equivariant function can be expressed, allowing a little different affine transformations. In particular, applying our results as a substitution action for $S_n$, the affine transformation from the input layer to the first hidden layer is different from $ aI + b11^T $. In that sense, it is different from the model of Zaheer et al. If there is an opportunity to modify, we will write down what affine transformations will be made and add them to the final version for clarity of comparison.
>
> > It was also not clear why it was necessary to present the contribution that with each symmetry you introduce you reduce the number of parameters exponentially? I guess the intuition behind this is clear, but I was wondering why this is necessary to spell out with a Theorem (2.3 I believe). What exactly doe this result imply?
>
> As you can see from the curse of dimensionality, the reduction of parameters is closely related to the number of samples required to perform the actual task, and neural networks with symmetry has fewer parameters than usual. Showing them gives the intuition that the required number of samples is reduced when we adapt it to the actual task. We are not yet able to mathematically formulate this, but we believe that the result of reducing the number of parameters is worth writing in the paper.

---

### Official Review · AnonReviewer1 · 2019-10-24
**Official Blind Review #1**

**Rating:** 3

**Review:**

The paper proposes a universal approximator for functions equivariant to finite group action. The idea is to draw a bijection between such equivariant universal approximators and those for functions that are “invariant” to the stabilizer subgroup of the output indices. Using existing results for designing universal invariant approximators, the paper then seems to suggest universal equivariant approximators in the form of neural networks.

While this is an important topic and the paper -- to the extent that I could follow -- seems to be technically sound, I found the paper very hard to read in part due to numerous grammatical errors.

Another issue is that I don’t see why the symmetric group is treated separately from the general finite groups. If in the end, the goal is to plug in “a” universal G-invariant function, the paper could leave those details out and focus on clarifying its proposed bijection. Could you please comment? Also is there a setting in which this setup leads to a practical architecture?


**Experience Assessment:**

I have published one or two papers in this area.

**Review Assessment: Checking Correctness Of Derivations And Theory:**

I did not assess the derivations or theory.

**Review Assessment: Checking Correctness Of Experiments:**

N/A

**Review Assessment: Thoroughness In Paper Reading:**

I read the paper at least twice and used my best judgement in assessing the paper.

---

> ### Author Response · Authors · 2019-11-14
> **Response to Review #1**
>
> We thank you for your definitive comments.
>
> The crucial point of this paper is exactly what you pointed out. Our original motivation was to prove a universal approximation theorem with DNN with affine transformation which commutative with group action. The proof of our universal approximation theorem for general finite groups is with very complicated notations, and is very similar to the proof for the symmetric group described in the paper.  So we decided to write around it.
> As you mentioned, the author understands that it is most important to be able to reduce the discussion of equivariant vector-valued functions to a discussion of few invariant functions. Therefore, we think that we should have described in detail how to write a $G$-equivariant function for a general finite group $G$ with an invariant function for some stabilizer groups. If allowed, we would like to revise the final version to emphasize it.

---

### Official Review · AnonReviewer2 · 2019-10-24
**Official Blind Review #2**

**Rating:** 3

**Review:**

This paper proposes a universal approximation theorem for functions invariant and equivariant to finite group actions. It constructs the approximations using fully-connected deep neural networks with ReLU activations. It proves a bound on the number of parameters in the build equivariant model. The proof structure uses a decompostion of G-equivariant functions into Stab(1)-invariant functions, which are represented by a particular network.

The proof structure is reasonably well written, however more context and examples could help a reader unfamiliar with the literature. This could be achieved by proposing applications of the construction of the approximating network to a few concrete applications where equivariant representations are needed.

As-is the paper is difficult for a newcomer to the field, compared e.g. with (Maron et al. 2019b) and (Keriven & Peyré 2019) which have clearer expositions. On the other hand, the equivariant representation problem is given in more generality, stemming from finite group actions represented by actions of permutation groups.

My current assessment is “borderline” and might change in light of author responses and reviews from reviewers with more experience in this field to judge the significance of the results.

I think it is necessary to do a more in-depth comparison with respect to the existing work of (Keriven & Peyré 2019).
In this regard, I think the following could be done:
- Giving an application and the corresponding construction where the increased generality w.r.t. the graph networks of (Keriven & Peyré 2019)
- In the case of a graph network already addressed by (Keriven & Peyré 2019) , how would this construction apply and how would the constructed network compare?

What do the authors call "usual models" in the discussion of Theorem 2.3? I assume this means models that do not exploit the equivariance of the function, but this could be more explicit.

Typos:
p.5, in the discussion of Theorem 2.1: "> Then, by ... we may assume"
p.8, Prop 4.1 Proof: n devides M
p.11: Prop A.1 Proof: can be realized as a permutation action on R^n: R should be in \mathbb
p.17: “ane” instead of “and”

***
Updated review

I have the impression all 3 reviewers have had some trouble going through the main messages of the paper. Although - at least in my case - this might be partly due to limited in-depth expertise, this seems to indicates some deeper shortcomings in the writing, as well as the insights and intuitions offered by the paper, in order to be accessible to a larger audience.

I still believe the approach may have merits, however I do not recommend acceptance of the paper at its current state. In my opinion, the following points should be considered for a future resubmission:
* A friendlier introduction to the matter with more intuitions and examples where the method has interest and distinguishes itself from existing papers, e.g. (Maron et al. 2019b), (Keriven & Peyré 2019) (Review #2, review #3)
* At least 1 example (and possibly more) where the proposed method leads to a practical architecture (Review #1)
* A clearer explanation of the particular role played by Sn w.r.t. general finite groups in the construction (Review #1)


**Experience Assessment:**

I do not know much about this area.

**Review Assessment: Checking Correctness Of Derivations And Theory:**

I assessed the sensibility of the derivations and theory.

**Review Assessment: Checking Correctness Of Experiments:**

N/A

**Review Assessment: Thoroughness In Paper Reading:**

N/A

---

> ### Author Response · Authors · 2019-11-14
> **Response to Review #2**
>
> We thank you for your constructive comments.
>
> > - Giving an application and the corresponding construction where the increased generality w.r.t. the graph networks of (Keriven & Peyré 2019)
> - In the case of a graph network already addressed by (Keriven & Peyré 2019) , how would this construction apply and how would the constructed network compare?
>
> They prove the universal approximation theorem by graph neural networks. This is because the adjacency matrix of the graph corresponds to rank 2 tensor which is an element of $R^{n ^ 2}$, and the isomorphism classes of the graph correspond to the equivalence classes with a special action of the symmetric group $S_n$. This results in the problem of the universal approximation theorem of invariant and equivariant functions for that action of $S_n$ on $R^{n ^ 2}$. On the other hand, we are giving results for more general actions on vector spaces. In particular, if we apply our results using their actions on tensors, we can obtain a universal approximation theorem as an array of networks of tensors with the action of stabilizer groups of $S_n$. In their results, higher order tensor $R^{n^k}$ with $S_n$-action needs to be considered in the middle layer. This corresponds to considering a hypergraph with edge size $k$. On the other hand, in our method, the result is obtained if the acting group is allowed to be small. We think that one of the important questions is how to interpret the equivalence class of $R^{n^2}$ in the action of the subgroup of $S_n$ as a graph.
> Also, it should be noted that the structure of the approximator is different from those of us. While they think of a kind of algebra and construct approximator using it, we regard equivariant vector-valued functions as an array of invariant functions and plug-in an approximator for invariant functions. As Reviewer #1 pointed out, this is an essential advantage of our results. Our method of construction also provides a framework for adapting what can be calculated for invariant functions to equivariant functions for other problems such as generalization errors, and is considered to be expansible.

---

### Decision · Program_Chairs · 2019-12-19

**Decision:**

Reject

**Comment:**

The article studies universal approximation for the restricted class of equivariant functions, which can have a smaller number of free parameters. The reviewers found the topic important and also that the approach has merits. However, they pointed out that the article is very hard to read and that more intuitions, a clearer comparison with existing work, and connections to practice would be important. The responses did clarify some of the differences to previous works. However, there was no revision addressing the main concerns.